# Decursin Induces G1 Cell Cycle Arrest and Apoptosis through Reactive Oxygen Species-Mediated Endoplasmic Reticulum Stress in Human Colorectal Cancer Cells in In Vitro and Xenograft Models

**DOI:** 10.3390/ijms25189939

**Published:** 2024-09-14

**Authors:** Danbee Kim, Seok-Ho Go, Yeeun Song, Dong-Keon Lee, Jeong-Ran Park

**Affiliations:** Division of Research Center, Scripps Korea Antibody Institute, Chuncheon 24341, Republic of Korea; kdb0515@skai.or.kr (D.K.); logos@skai.or.kr (S.-H.G.); yenixen@skai.or.kr (Y.S.)

**Keywords:** decursin, apoptosis, endoplasmic reticulum stress, reactive oxygen species, colorectal cancer

## Abstract

Decursin, a coumarin isolated from *Angelica gigas* Nakai, possesses anti-inflammatory and anti-cancer properties. However, the molecular mechanisms underlying its anti-cancer effects against human colorectal cancer (CRC) are unclear. Therefore, this study aimed to evaluate the biological activities of decursin in CRC in vitro and in vivo and to determine its underlying mechanism of action. Decursin exhibited anti-tumor activity in vitro, accompanied by an increase in G1 cell cycle arrest and apoptosis in HCT-116 and HCT-8 CRC cells. Decursin also induced the production of reactive oxygen species (ROS), thereby activating the endoplasmic reticulum (ER) stress apoptotic pathway in CRC cells. Furthermore, the role of ROS in decursin-induced apoptosis was investigated using the antioxidant N-acetyl-L-cysteine. Inhibiting ROS production reversed decursin-induced ER stress. Moreover, decursin significantly suppressed tumor growth in a subcutaneous xenograft mouse model of HCT-116 and HCT-8 CRC cells without causing host toxicity. Decursin also decreased cell proliferation, as documented by Ki-67, and partly increased cleaved caspase 3 expression in tumor tissues by activating ER stress apoptotic pathways. These findings suggest that decursin induces cell cycle arrest and apoptosis in human CRC cells via ROS-mediated ER stress, suggesting that decursin could be a therapeutic agent for CRC.

## 1. Introduction

Colorectal cancer (CRC) is a common gastrointestinal tract malignancy. CRC has the third-highest incidence and ranks second in terms of mortality among all malignancies [1]. Therefore, reducing the incidence and mortality rates of CRC is an urgent clinical issue. Current CRC treatments include surgery, radiotherapy, and chemotherapy [2,3,4,5]. However, common chemotherapeutic agents such as oxaliplatin, irinotecan, and 5-fluorouracil (5-FU) cause undesirable side effects. 5-FU is the first drug of choice for CRC treatment. However, the overall response rate of 5-FU in CRC is only approximately 10–15%, even when used in combination with other chemotherapeutic agents [6,7,8]. 5-FU also has side effects such as nausea, diarrhea, headache, skin itching, and anemia [9]. Chemotherapy also has various adverse events, and chemotherapeutic resistance is a major concern in clinical applications. Therefore, identifying new anti-tumor drugs that exhibit low toxicity with less potential for the development of drug resistance is essential.

Phytochemicals are plant-based bioactive compounds produced from fruits, vegetables, and medicinal plants. Phytochemicals have recently been investigated for cancer treatment and prevention [10,11] because of their low cost, low intrinsic toxicity, few side effects, and potent anti-cancer activity [12,13,14]. Many phytochemicals suppress proliferation by inducing cell cycle arrest and apoptosis in CRC cells, thereby inhibiting tumor growth [15,16]. In recent years, phytochemicals have been extensively screened to investigate their potential anti-tumor activity. A variety of FDA-approved anti-tumor agents, including docetaxel and topotecan, have demonstrated clinical utility based on ongoing investigations of natural products.

Decursin is a bioactive phytochemical originally isolated from *Angelica gigas* Nakai (AGN) and is one of the most popular traditional medicines in East Asian countries. It has been used in traditional folk medicine as a tonic and a remedy for anemia, colds, pain, and other ailments; AGN is also marketed internationally as a functional food for healthcare [17,18]. Decursin has been investigated to have a variety of therapeutic effects in increasing numbers of studies, such as anti-angiogenesis, anti-oxidative, and anti-inflammatory activities [19,20,21,22]. Notably, decursin has exhibited anti-cancer properties in multiple cancer types, such as prostate, breast, lung, bladder, and colon [20,23,24,25]. Some reports have demonstrated the effects of decursin on CRC. Son et al. showed that decursin suppress the proliferation and invasion of murine CT-26 colon carcinoma cells [26]. Kim et al. demonstrated that decursin inhibits cell growth in 253J bladder and HCT-116 colon cancer cells via cell cycle arrest and ERK activation in vitro [25]. Most recent reports have shown that decursin regulates epithelial–mesenchymal transition (EMT) via the PI3K/Akt signaling pathway in CRC cells [27]. These effects of decursin make it an attractive candidate for further investigation and development as a CRC treatment. However, the detailed molecular mechanisms underlying the anti-cancer activity of decursin in CRC cells remain largely unexplored. Additionally, little is known about the systemic exposure of decursin following administration, as its anti-cancer activity has not been well explored in vivo compared to its in vitro effects.

The present study represents the first comprehensive examination of the anti-cancer efficacy of decursin in the treatment of CRC cells, in both in vitro and xenograft models, while also elucidating the underlying molecular mechanisms. In addition, this study aimed to investigate the biological activities of decursin in CRC using both in vitro and in vivo models, as well as to determine its underlying mechanism of action, providing scientific evidence regarding the therapeutic roles of decursin.

## 2. Results

### 2.1. Decursin Inhibits Crc Cell Growth

To investigate the effects of decursin on the cell growth of CRC cells in vitro, the viability rate of HCT-116 and HCT-8 CRC cells was assessed using an MTT assay. Treatment with decursin inhibited cell viability in a dose- and time-dependent manner at different concentrations (3.125, 6.25, 12.5, 25, 50, 100, and 200 µM) for 24 to 72 h compared with control cells (Figure 1A). In addition, the half-maximal inhibitory concentration (IC_50_) value for 72 h of decursin treatment was approximately 35 µM in HCT-116 and 30 µM in HCT-8 CRC cells (Figure 1B). These values indicate that HCT-116 and HCT-8 CRC cells are sensitive to decursin treatment. Based on the IC_50_ measured in the study, concentrations of 50 and 100 μM were selected for use in the following experiments. Decursin also induced significant morphological changes in the CRC cell lines (Figure 1C). In the treatment group, the cell bodies became smaller and rounder than those in the control group. Additionally, significant cellular debris appeared in the culture medium, and the cell count decreased. These results show that decursin suppresses the growth of HCT-116 and HCT-8 CRC cells.

### 2.2. Decursin Triggers Cell Cycle Arrest at the G1 Phase in Crc Cells

A significant increase in the number of cells that were arrested in the G1 phase was observed, and the proportion of cells in the S phase decreased in a concentration-dependent manner (Figure 2A,B). Furthermore, the protein levels associated with the G1 phase and cycles D1 and CDK4 were downregulated after decursin treatment. Similarly, proteins related to the S and G2/M phases and cycles E and CDK2 were also downregulated. In addition, p21 expression, a CDK inhibitory protein that suppresses CDK activation, continuously increased (Figure 2C). These results suggest that decursin induced G1 phase arrest and S phase delay.

### 2.3. Decursin Induces Apoptosis in Crc Cells

Annexin-V fluorescein isothiocyanate/propidium iodide (FITC/PI) staining showed that the proportions of apoptotic cells were markedly increased in a dose-dependent manner in HCT-116 and HCT-8 CRC cells when CRC cells were incubated with concentrations of 50 and 100 μM decursin (Figure 3A,B). The percentage of apoptotic HCT-116 (HCT-8) cells after 48 h treatment with 0, 50, and 100 μM decursin were 3.82% (2.94%), 20.5% (14.06%), and 40.44% (43.08%), respectively. Furthermore, Western blotting showed that the expression levels of cleaved caspase 3, Bax, and cleaved PARP were significantly increased after decursin treatment, whereas the expression level of Bcl-xL was decreased in HCT-116 and HCT-8 CRC cells (Figure 3C). These results suggest that the inhibition of cell growth by decursin was associated with the induction of apoptosis. 

### 2.4. Decursin Causes Reactive Oxygen Species Accumulation and Endoplasmic Reticulum Stress in Crc Cells

Treating HCT-116 and HCT-8 CRC cells with decursin caused a significant increase in reactive oxygen species (ROS) production in a dose-dependent manner (Figure 4A,B). The expression levels of the components of the endoplasmic reticulum (ER) stress pathway, such as G protein-coupled receptor 78 (GRP78), PKR-like ER kinase (p-PERK), p-α subunit of eukaryotic initiation factor 2 (p-eIF2a), transcription factor 4 (ATF4), and C/EBP-homologous protein (CHOP), were significantly increased in a dose-dependent manner. These results indicate that decursin can significantly activate ER stress. When cells were pretreated with 4-phenylbutyric acid (4-PBA) and decursin, the cell survival rate increased compared to that of cells treated with decursin alone (Figure 4D). These results suggest that ER stress may be a major factor in the decursin-induced apoptosis of CRC cells.

### 2.5. Decursin Induces Ros-Mediated Cell Cycle Arrest and Apoptosis via ER Stress in Crc Cells

After treatment with N-acetyl-L-cysteine (NAC), intracellular ROS production decreased significantly compared to that in cells treated with decursin alone (Figure 5A,B). Furthermore, NAC treatment significantly increased the cell viability compared to that of cells treated with decursin alone (Figure 5C). Western blotting showed that NAC treatment significantly decreased the expression of ER stress-related proteins, including p-PERK, p-eIF2a, and CHOP, in HCT-116 and HCT-8 CRC cells compared with decursin treatment alone (Figure 5D). These results suggest that decursin induces ER stress in CRC cells through ROS accumulation and that ER stress participates in the decursin-mediated inhibition of cell proliferation, cell cycle arrest, and apoptosis.

### 2.6. Decursin Suppresses Tumor Growth of Crc Cells In Vivo

The tumor tissues in BALB/c nude mice were excised (Figure 6A,E), and the tumor volume (Figure 6B,F) and body weight (Figure 6C,G) were evaluated. Throughout treatment, the decursin-treated group showed a significant delay in tumor growth (% TGI of HCT-116 = 61.1%; TGI of HCT-8 = 54.3%) compared to that of the control group (Figure 6D,H). In addition, no significant difference in body weight was observed between the control and decursin-treated groups. Hematoxylin and eosin (H&E) staining revealed a significant increase in tumor necrosis in the mice treated with decursin (Figure 6Ia,Ja,K,L). Furthermore, decursin decreased Ki-67 expression (Figure 6Ib,Jb), and more cleaved caspase 3-positive cells were observed in the decursin-treated group than in the control group (Figure 6Ic,Jc). Western blotting also suggested that decursin treatment increased the expression of apoptosis-inducing proteins, such as cleaved caspase 3 and PARP, and ER stress-related proteins, especially CHOP, and significantly decreased cell cycle-related proteins in tumor xenografts (Figure 7). These results indicate that decursin inhibited cell proliferation and induced apoptosis via ER stress pathways in a human colorectal tumor xenograft model, consistent with the in vitro results.

## 3. Discussion

CRC is the third most common malignancy and one of the leading causes of cancer-related deaths worldwide [28]. However, chemotherapy is limited by drug resistance and a narrow mechanism of action, and the overall survival rate of patients with advanced CRC remains low [29,30]. Therefore, more effective and less toxic therapeutic agents are required for the treatment of CRC to increase the overall survival rate and reduce the side effects. Natural medicinal herbs and plant-based phytochemicals have played essential roles as effective sources of anti-tumor effects and have potential against various cancers by regulating selective molecular targets [31,32,33]. Phytochemicals are becoming increasingly vital in drug discovery because of their high molecular diversity and novel biofunctionalities. Decursin has potential antioxidant, anti-inflammatory, and anti-cancer effects by modulating specific signaling pathways in some tumor models [21,25,34,35]. 

This study demonstrated that decursin exerted anti-tumor effects against CRC cells by inhibiting cell proliferation and tumor growth in vitro and in vivo. Furthermore, decursin induced apoptosis via ROS-mediated ER stress in CRC cells. Decursin also induced G1 cell cycle arrest and inhibited cyclins (cyclin D1 and E) and cyclin-dependent kinases (CDK2 and CDK4), which are vital regulators of cell cycle progression in HCT-116 and HCT-8 CRC cells. p21 typically binds to the CDK complex, inhibiting its kinase activity and preventing cell cycle progression during the G1 and S phases [36,37]. In this study, decursin upregulated p21 expression but downregulated cyclin D1 and E expression. These results suggest that decursin induces cell cycle arrest and that decursin-induced G1 cell cycle arrest provides an opportunity for CRC cells to undergo apoptotic progression in CRC cells. 

Most of the presently available anti-cancer drugs have the ability to induce tumor cell apoptosis. In various cancer cells, the anti-cancer effects of decursin have been documented—gastric cancer cells (IC_50_ = 50 µM for 48 h) [38], bladder cancer cells (IC_50_ = 50 µM for 24 h) [25], melanoma (IC_50_ = 80 µM for 48 h) [21], and multiple myeloma cells (IC_50_ = 80 µM for 24 h–48 h) [39]. In the present study, decursin was also able to effectively inhibit the cell growth of HCT-116 and HCT-8 CRC cells at 48 h, with IC_50_ values of 50.33 µM and 49.68 µM, respectively. Apoptosis is an essential form of cell death that is closely related to tumor development and treatment. PARP is a crucial enzyme involved in DNA repair, and its cleavage by caspases is considered a hallmark of apoptosis [40,41]. Bax and Bcl-xL are pro-apoptotic and anti-apoptotic members of the Bcl-2 family, respectively, and play essential roles in mitochondria-mediated apoptosis [42,43]. In this study, decursin significantly upregulated cleaved PARP and Bax expression but downregulated Bcl-xL expression. Because caspase 3 plays a crucial role in apoptosis, the changes in cleaved caspase 3 were further investigated [44]. The expression level of cleaved caspase 3 in decursin-treated CRC cells was significantly lower than that in control cells. These results suggest that decursin induces apoptosis in CRC cells via the mitochondrial apoptotic pathway.

A clearer understanding of how decursin regulates the cell death pathway could provide crucial insights into its underlying mechanisms. ROS production and ER stress are essential pathways involved in cell proliferation and apoptosis, and decursin induces apoptosis via ROS-mediated ER stress signaling in non-small-cell lung cancer [23]. ROS generation is an essential factor in tumor cell death [45], and ROS production promotes cell cycle arrest induced by multiple anti-cancer drugs [46]. In this study, NAC dramatically reversed the effects of decursin on the viability of HCT-116 and HCT-8 CRC cells. These results suggest that decursin exerts apoptotic effects by producing ROS in CRC cells. 

Increased ROS levels and the perturbation of the intracellular redox status increase the levels of unfolded proteins in the ER and induce an ER stress response [47]. Under oxidative stress conditions, ER stress plays a vital role in inducing apoptosis in various tumors, including breast cancer, prostate cancer, and melanoma [48,49,50]. CHOP is a crucial regulator of ER stress-induced cell death [51]. In this study, decursin induced ER stress by upregulating GPR78, PERK, and CHOP expression in CRC cells. The initiation of the ER stress-induced apoptotic pathway increases CHOP gene expression, triggering an ER stress-specific cascade for the implementation of apoptosis. Consequently, in this study, the upregulation of CHOP expression was observed in CRC cells after decursin treatment. Additionally, 4-PBA reversed cell growth after decursin treatment. Moreover, the decursin-induced activation of ER stress was inhibited by NAC, indicating that ROS act as upstream signaling molecules involved in decursin-induced ER stress in CRC cells. This finding suggests that decursin induces apoptosis by activating ROS-mediated ER stress in CRC cells (Figure 8).

While in vitro cell culture models are an effective system for initially screening the effects of chemotherapeutic agents, validating these findings in vivo through animal studies is essential before considering their potential use in humans. In this study, decursin treatment inhibited the growth of HCT-116 and HCT-8 tumor xenografts without any apparent signs of toxicity in mice. These results were in accordance with the decreased proliferation documented by Ki-67 immunostaining. In addition, decursin partly induced apoptosis by activating the ER stress pathway, as indicated by cleaved caspase 3 expression in tumor tissues. These in vivo results are consistent with those of the in vitro study, suggesting that decursin could be a practical approach to enhance the anti-tumor activity against CRC. 

Phytochemicals are the primary source of biologically active compounds. Moreover, their nontoxic or less toxic nature to normal cells has gained attention from the scientific community and clinicians in the modern drug discovery field [52]. Several studies have examined the cytotoxicity of decursin in normal cells. Up to 100 µg/mL of AGN extract and 60 µM of decursin inhibit the growth of PANC-1 and MIA PaCa-2 pancreatic cancer cells, but not normal pancreatic cells [53]. Decursin at 80 µM exhibited an apoptotic effect in human KBM-5 leukemia cells but did not show cytotoxicity against human peripheral blood lymphocytes at concentrations up to 80 µM [54]. Furthermore, in non-neoplastic human prostate epithelial PWR-1E cells, decursin treatment did not lead to any increase in dead cells after 72 h of treatment at a dose of 100 μM [55]. Additionally, Kim et al. evaluated the acute and subacute effects of decursin in SD rats to determine its effects on body weight, as well as histopathological, biochemical, and hematochemical changes. There were no significant differences in body weight, hematology, and biochemical parameters between the control and treated groups [56]. These results suggest that decursin may be an effective and safe treatment option for CRC therapy.

Phytochemicals have been used for the treatment and prevention of various diseases, either alone or in combination with other drugs [57,58]. Therefore, we have explored the potential therapeutic effects of decursin in combination with oxaliplatin, a known active agent in first-line chemotherapy for advanced CRC. The results indicate that decursin sensitizes anti-cancer agents, and further studies are currently ongoing. Additional detailed preclinical research is required to identify and discover more effective synergistic combinations of decursin with alternative medications to prevent the failure of CRC therapy.

## 4. Materials and Methods

### 4.1. Cell Culture and Decursin Solution Preparation

HCT-116 and HCT-8 human CRC cell lines were purchased from the Korean Cell Line Bank (KCLB, Seoul, Republic of Korea). Both cell lines were cultured in RPMI-1640 medium (Hyclone, Logan, UT, USA) with 10% fetal bovine serum and 1% penicillin–streptomycin in a 5% CO_2_ incubator maintained at a temperature of 37 °C. 

Decursin (99.99% purity, Selleckchem, Inc., Houston, TX, USA) was dissolved in DMSO to prepare a stock solution. The final concentration of DMSO in treatment solutions was less than 0.1%.

### 4.2. Cell Viability Assay

The cells were seeded at 1 × 10^4^ cells/well in 96-well plates and were incubated overnight. Cells were treated with various decursin concentrations (0, 12.5, 25, 50, 100, and 200 μM) for 48 h. Then, 100 uL of MTT (a 3-(4,5-dimethylthiazol-2-yl)-2,5-diphenyltetrazolium bromide) at a dose of 5 mg/mL (Sigma-Aldrich, St. Louis, MO, USA) was added to each well and was further incubated for 2 h at 37 °C. The supernatant was removed, and 200 uL of dimethyl sulfoxide was added to the wells to dissolve the purple formazan crystals. Quantification was performed by measuring the absorbance at 540 nm. Results are presented as the mean values from three independent experiments performed in triplicate. The IC_50_ values were calculated using dose–response curves and GraphPad Prism 5 (GraphPad Software Inc., La Jolla, CA, USA).

### 4.3. Cell Cycle Analysis

Cells were treated with different decursin concentrations (0, 50, or 100 μM) for 48 h and then harvested using trypsinization. After washing the cells with ice-cold phosphate-buffered saline (PBS), 70% ethanol was added to fix the cells at 4 °C for 4 h. Fixed cells were stained with FxCycle propidium iodide/RNase Staining Solution (Invitrogen, Waltham, MA, USA) for 30 min at room temperature. Cell cycle distribution was analyzed using flow cytometry (Agilent NovoCyte, Santa Clara, CA, USA).

### 4.4. Cell Apoptosis Assay

Cells were treated with decursin at concentrations of 0, 50, and 100 μM for 48 h, and were harvested using the Annexin-V FITC/PI apoptosis detection kit according to the manufacturer’s instructions (BD, Franklin Lakes, NJ, USA). Samples were assayed using flow cytometry (Agilent NovoCyte, Santa Clara, CA, USA).

### 4.5. Intracellular ROS Measurement

2,7-Dichlorofluorescein-diacetate (DCF-DA) was used to determine the intracellular ROS levels. Cells were loaded with 20 μM DCF-DA (Abcam, Cambridge, MA, USA) for 30 min at 37 °C, followed by treatment with various decursin concentrations for 48 h. The cells were detached using 0.25% trypsin and analyzed using flow cytometry (Agilent NovoCyte, Santa Clara, CA, USA). To further confirm that ER stress plays a vital role in decursin-induced CRC cell apoptosis, cells were pretreated with the ER stress inhibitor 4-PBA for 1 h before treatment with decursin. Lastly, to investigate the relationship between ROS production and decursin-induced cell death in CRC cells, NAC was used to scavenge ROS, and its effects on HCT-116 and HCT-8 CRC cells were determined.

### 4.6. Western Blotting

Whole-cell lysates were prepared with RIPA lysis buffer containing a protease/phosphatase inhibitor cocktail (Cell Signaling Technology, Danvers, MA, USA) on ice, and 30 μg of isolated proteins were separated on 4–20% sodium dodecyl-sulfate-polyacrylamide gels. The separated proteins were transferred to polyvinylidene difluoride membranes. The membranes were incubated with 5% skim milk for 1 h and then probed with the following 1:500–1:1000 diluted antibodies to detect the respective proteins at 4 °C overnight: anti-caspase-3, anti-cleaved caspase-3, anti-PARP, anti-cleaved PARP, anti-Bax, anti-Bcl-xL, anti-cyclin D, anti-cyclin E, anti-CDK4, anti-CDK2, anti-p21 Waf1/Cip1, anti-ATF4, anti-CHOP, anti-eIF2α, anti-peIF2α (Cell Signaling Technology), anti-PERK, anti-β-actin (Santa Cruz Biotechnology, Santa Cruz, CA, USA), anti-pPERK, and anti-BiP/GRP78 (Abclonal, Woburn, MO, USA). After four washes for 5 min each, polyclonal anti-rabbit/mouse horseradish peroxidase (HRP)-conjugated secondary antibodies were linked to HRP-bound protein complexes and developed using Pierce ECL Western Blot substrate (Thermo Fisher Scientific, Waltham, MA, USA). The protein bands were imaged using an iBright Imaging System (Invitrogen). Band densitometry was performed using ImageJ2 software (National Institutes of Health, Bethesda, MD, USA) and was expressed as fold-change relative to β-actin.

### 4.7. Animal Studies

All animal experimental procedures were approved by the Institutional Animal Care and Use Committee of the Scripps Korea Antibody Research Institute (approval number: SKAI-221116-1). For the mouse xenograft model, a suspension of 5 × 10^6^ HCT-116 or HCT-8 cells dissolved in 0.1 mL PBS was inoculated subcutaneously into the right flank of 6-week-old female BALB/c nude mice (Koatech, Gyeongi-do, Republic of Korea). When the tumors reached approximately 75–100 mm^3^, the mice were randomly divided into a control (PBS) and a decursin group (10 mg/kg), with five mice per group. The administered substance was injected intraperitoneally twice weekly for 14 days. Body weight and tumor volume were measured every two days. The tumor volume was calculated using the following formula: (length × width ×width)/2. All mice were anesthetized using isoflurane inhalation (Hana Pharma, Gyeonggi-do, Republic of Korea) and euthanized with CO_2_ at the end of the experiment, after which the tumors were harvested for further analysis. 

### 4.8. H&E Staining and Immunohistochemistry

H&E staining was performed on xenograft tumors. Briefly, the freshly dissected tissues were fixed and embedded in paraffin. After being cut into 5 μm slices, the sections were deparaffinized in xylene and stained in Mayer’s H&E solution. Finally, the sections were dehydrated and mounted using Permount in a fume hood. 

The uncontrolled proliferation of tumor cells is a characteristic feature of most cancers. Therefore, tumor xenografts were analyzed for the potential anti-proliferative effects of decursin using immunohistochemistry (IHC) of cell proliferation marker Ki-67-positive cells. Additionally, the apoptotic effect of decursin on tumor tissues was assessed by evaluating cleaved caspase-3 expression, a marker of apoptosis, to determine whether the inhibition of tumor growth following decursin treatment was due to the apoptosis of tumor cells in xenograft tissues. For IHC, the freshly dissected tumor tissue was fixed in 10% neutral-buffered formalin for 48 h. The tissue was embedded in paraffin, cut to a thickness of 5 μm using a microtome, and affixed onto a slide. Tumor sections were deparaffinized in xylene and hydrated in ethanol, and antigen retrieval was performed using citrate buffer (pH 6.0) (Sigma-Aldrich, St. Louis, MO, USA). After that, the tumor sections were blocked and incubated with primary antibodies against Ki-67 and cleaved caspase 3 (Cell Signaling Technology, Danvers, MA, USA) overnight at 4 °C, as well as HRP-conjugated IHC detection reagent (Cell Signaling Technology, Danvers, MA, USA) for 30 min at room temperature. Then, 3,3′-diaminobenzidine was used to visualize the IHC results. The sections were counterstained with hematoxylin for 2–3 min, mounted, and analyzed under a light microscope (Leica, Deerfield, IL, USA).

### 4.9. Statistical Analysis

All data are presented as the mean ± standard deviation unless otherwise indicated. The statistical significance of the differences between the values of the control and treatment groups was analyzed using Student’s t-test or one-way analysis of variance, followed by Tukey’s post hoc test for multiple comparisons using GraphPad Prism 5 (GraphPad Software). *p*-values of <0.05 were considered statistically significant.

## 5. Conclusions

The findings of this study demonstrate that decursin activates the ROS-dependent ER stress apoptotic pathway and exhibits strong anti-cancer activity against CRC by inducing apoptosis and inhibiting cell growth. Therefore, decursin could be a promising bioactive phytochemical for CRC treatment. 

## Figures and Tables

**Figure 1 ijms-25-09939-f001:**
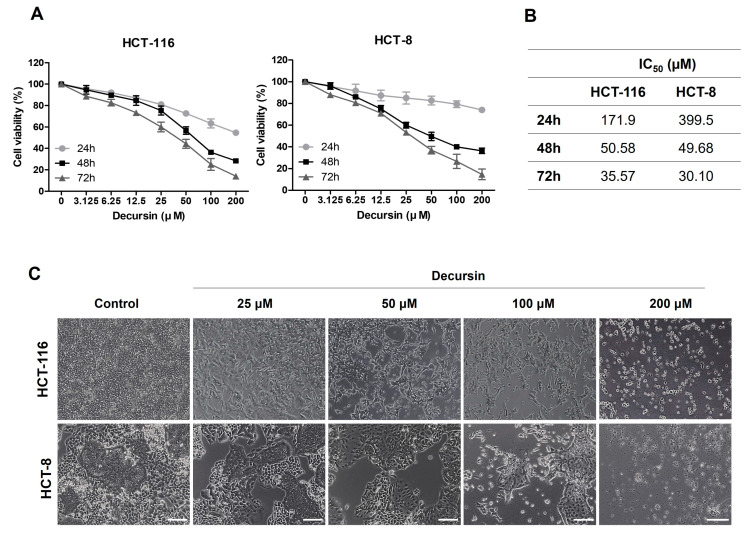
Decursin decreases cell viability in colorectal cancer cells. (**A**) Cell viability of HCT-116 and HCT-8 colorectal cancer (CRC) cells after treatment with decursin for 24, 48, and 72 h was detected using an MTT assay. (**B**) The half-maximal inhibitory concentration of decursin in CRC cells. (**C**) Morphology of HCT-116 and HCT-8 CRC cells after treatment with the indicated decursin concentrations for 48 h. Data are represented as the mean ± standard deviation (SD) of three independent experiments.

**Figure 2 ijms-25-09939-f002:**
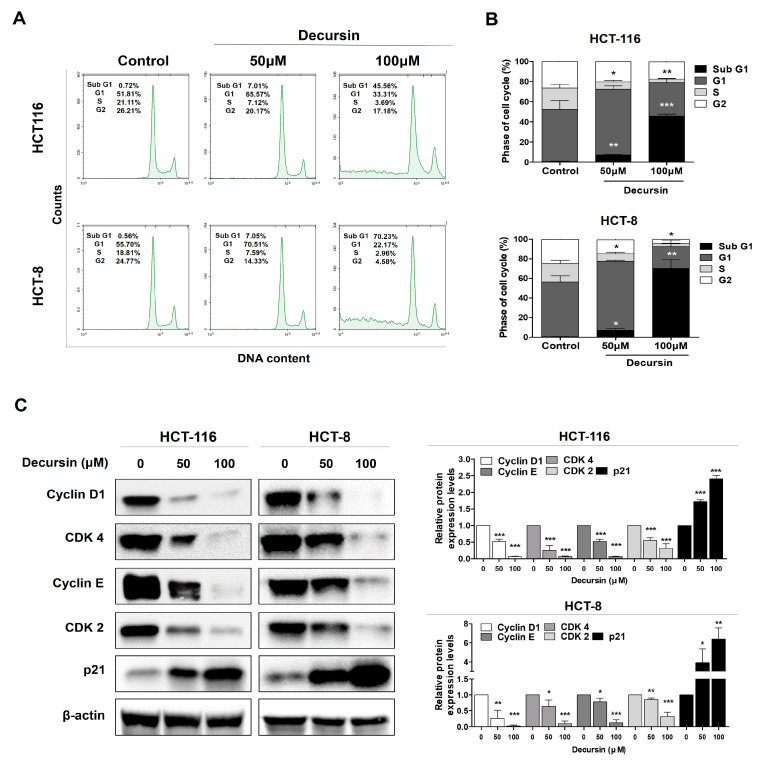
Decursin causes cell cycle arrest in CRC cells. (**A**,**B**) HCT-116 and HCT-8 CRC cells treated with decursin were stained with Annexin-V fluorescein isothiocyanate/propidium iodide (FITC/PI) and analyzed using flow cytometry. (**C**) Cyclin D1, CDK4, cyclin E, CDK2, and P21 protein expression levels in Western blotting. β-actin was used as an internal control. Data are represented as the mean ± SD of three independent experiments. * *p* < 0.05, ** *p* < 0.01, and *** *p* < 0.001 compared to the control group.

**Figure 3 ijms-25-09939-f003:**
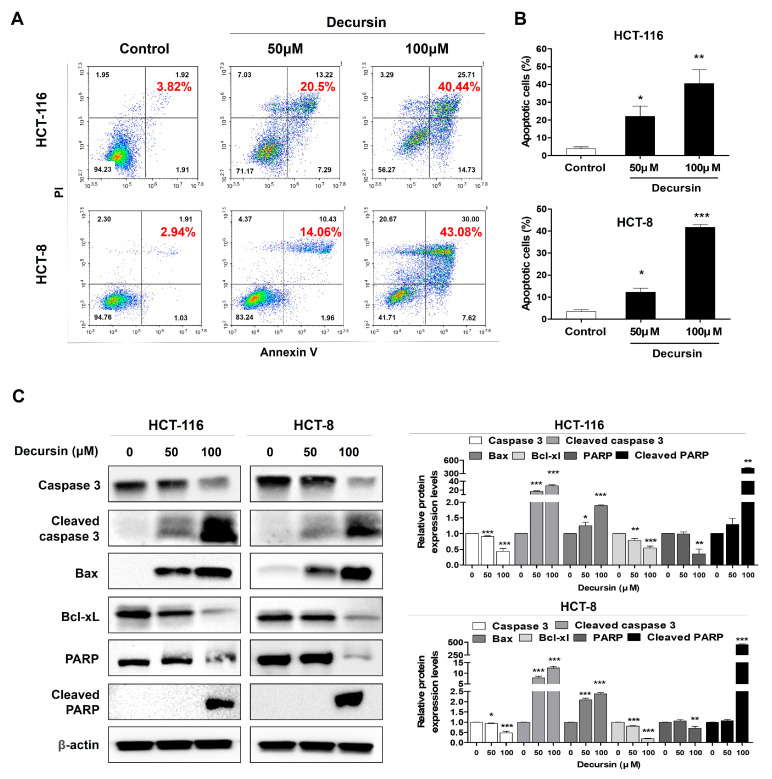
Decursin induces apoptosis in CRC cells. (**A**) Decursin-induced apoptosis was measured using an Annexin-V FITC/PI apoptosis detection kit and flow cytometry. (**B**) The percentage of apoptotic cells in the treatment group was calculated. (**C**) HCT-116 and HCT-8 CRC cells were treated with 0, 50, and 100 μM decursin for 48 h. Cell lysates were subjected to Western blotting to assess the expression of cell apoptosis-related proteins, including PARP, cleaved PARP, caspase-3, cleaved caspase-3, Bax, and Bcl-xL. β-actin was used as an internal control. Data are represented as the mean ± SD of three independent experiments. * *p* < 0.05, ** *p* < 0.01, and *** *p* < 0.001 compared to the control group.

**Figure 4 ijms-25-09939-f004:**
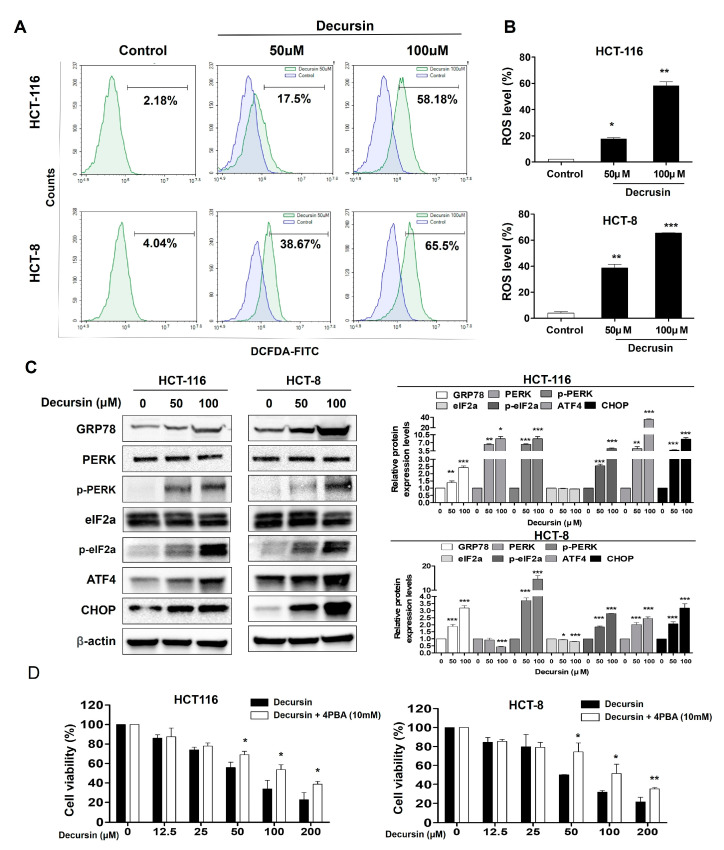
Decursin causes reactive oxygen species accumulation and endoplasmic reticulum stress in CRC cells. (**A**) Intracellular reactive oxygen species (ROS) levels were measured using flow cytometry. The *x*-axis represents the intensity of intracellular 2,7-dichlorofluorescein-diacetate fluorescence, and the *y*-axis indicates the mean number of cells. (**B**) Bar graphs showing that decursin led to an increased ROS level. (**C**) Cells were treated with decursin for 48 h, and the G protein-coupled receptor 78 (GRP78), PERK, PKR-like ER kinase (p-PERK), eIF2a, p-α subunit of eukaryotic initiation factor 2 (p-eIF2a), transcription factor 4 (ATF4), and C/EBP-homologous protein (CHOP) protein expression levels were determined using Western blotting. β-actin was used as an internal control. (**D**) Pretreatment with 10 mM 4-phenylbutyric acid for 1 h was followed by decursin treatment. Cell viability was measured using the MTT assay. Data are represented as the mean ± SD of three independent experiments. * *p* < 0.05, ** *p* < 0.01, and *** *p* < 0.001 compared to the control group.

**Figure 5 ijms-25-09939-f005:**
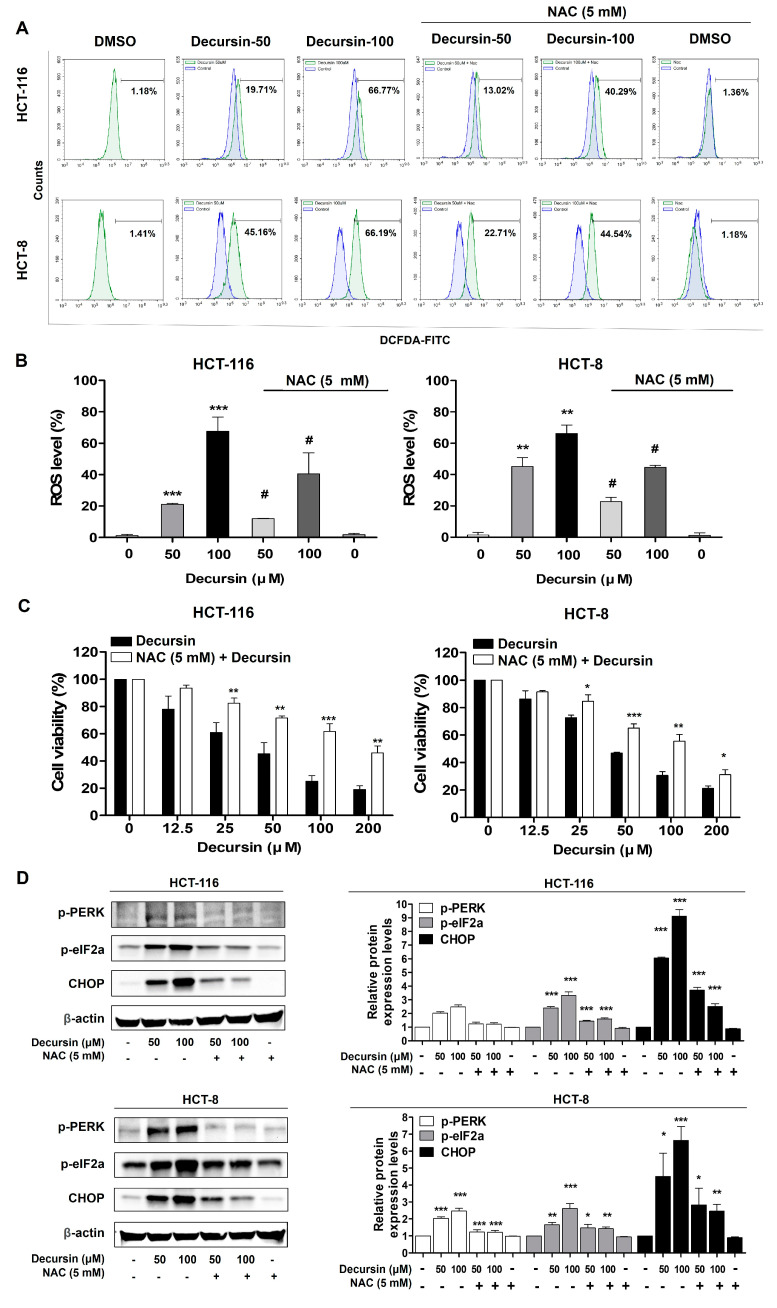
Decursin induces ER stress via ROS production in CRC cells. (**A**) Flow cytometry was performed to determine the intracellular ROS levels in HCT-116 and HCT-8 CRC cells with or without 5 mM N-acetyl-L-cysteine (NAC) treatment for 1 h before decursin administration for 48 h. (**B**) The bar graph shows that 5 mM NAC treatment reduces decursin-induced ROS accumulation. (**C**) An MTT assay was performed to determine the cell viability of HCT-116 and HCT-8 CRC cells with or without 5 mM NAC treatment for 1 h before decursin treatment for 48 h (n = 3 for each group). (**D**) HCT-116 and HCT-8 CRC cells were pretreated with 5 mM NAC for 1 h, then incubated with decursin for 48 h. Western blotting was used to determine p-PERK, p-elF2a, and CHOP expression. β-actin was used as an internal control. Data are represented as the mean ± SD of three independent experiments. * *p* < 0.05, ** *p* < 0.01, and *** *p* < 0.001 compared to the control group; ^#^
*p* < 0.05 decursin vs. decursin + NAC.

**Figure 6 ijms-25-09939-f006:**
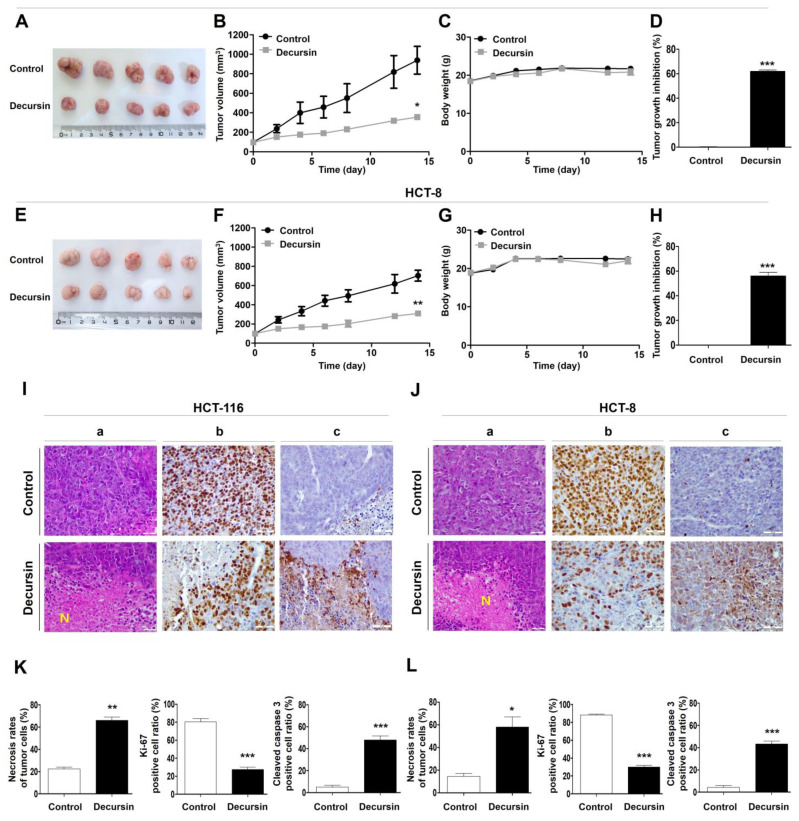
Decursin inhibits the growth of mouse xenograft CRC cells in vivo. The HCT-116 and HCT-8 cell xenograft mouse models were treated with phosphate-buffered saline and were intraperitoneally injected with decursin (10 mg/kg) twice weekly for 15 days. Representative images of (**A**) HCT-116 and (**E**) HCT-8 implanted tumors in each group at study termination. (**B**,**F**) Tumor volumes were measured on the indicated treatment days. (**C**,**G**) Mice body weight. (**D**,**H**) Tumor growth inhibition rate. (**I**–**L**) Immunohistochemistry (IHC) analysis. Representative images of IHC and (**Ia**,**Ja**) hematoxylin and eosin staining (**Ib**,**Jb**) for Ki-67 and (**Ic**,**Jc**) cleaved caspase 3 expression in tumor tissues. (**K**–**L**) For the staining analysis of necrotic rate/Ki-67/cleaved caspase 3 in tumor tissue, the percentage of positive cells (×400) among > 100 cancer cells was calculated in three randomly selected fields of view using a high-power lens (five mice per group, * *p* < 0.05, ** *p* < 0.01, and *** *p* < 0.001 compared to the control group).

**Figure 7 ijms-25-09939-f007:**
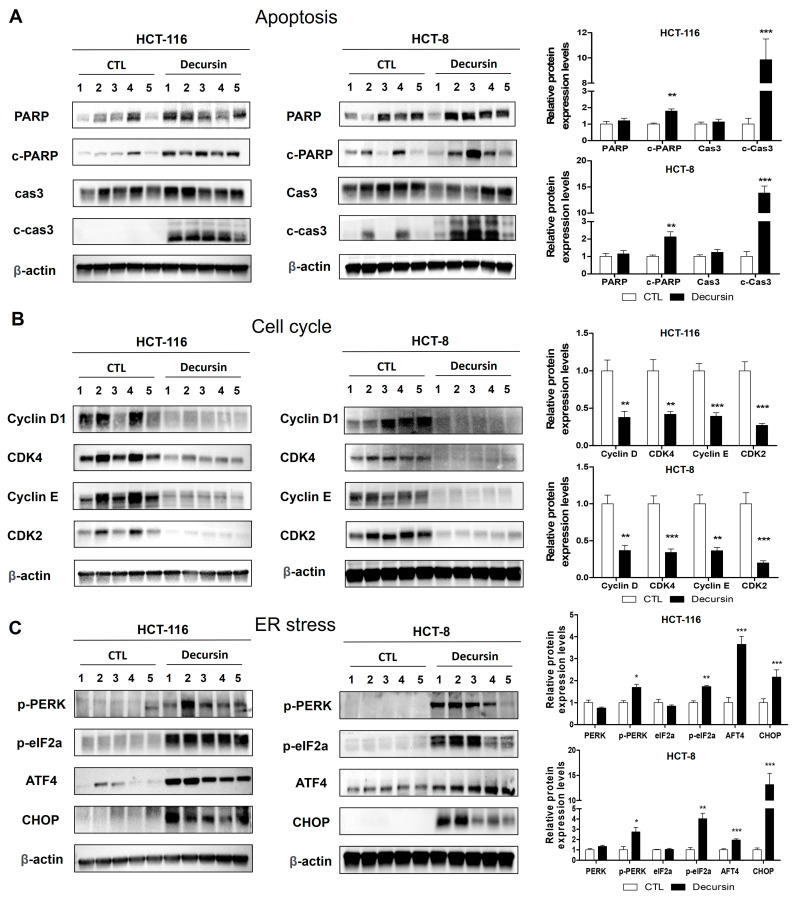
Decursin induces apoptosis, cell cycle arrest, and ER stress in mouse xenograft CRC in vivo. The patterns of changes in proteins related to apoptosis, the cell cycle, and ER stress in tumor tissues in vivo were analyzed using Western blotting. Tumor tissue lysates were immunoblotted using antibodies against (**A**) apoptosis-related proteins—PARP, cleaved PARP, caspase-3, and cleaved caspase-3; (**B**) cell cycle-related proteins—cyclin D, CDK4, cyclin E, and CDK2; and (**C**) ER stress-related proteins—p-PERK, p-eIF2α, ATF4, and CHOP. β-actin was used as a loading control (five mice per group; * *p* < 0.05, ** *p* < 0.01, and *** *p* < 0.001 compared to the control group).

**Figure 8 ijms-25-09939-f008:**
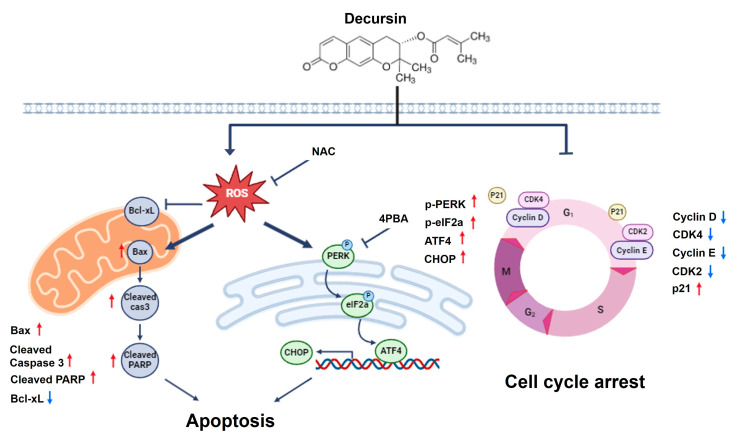
Schematic diagram of the proposed model system. Proposed model of the mechanism underlying the anti-tumor effects of decursin in HCT-116 and HCT-8 CRC cells. Decursin induces cell death by activating the ROS-mediated ER stress pathway.

## Data Availability

The datasets used and/or analyzed during the current study are available from the corresponding author on reasonable request.

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
