# Peer review of "Decursin Induces G1 Cell Cycle Arrest and Apoptosis through Reactive Oxygen Species-Mediated Endoplasmic Reticulum Stress in Human Colorectal Cancer Cells in In Vitro and Xenograft Models"

_ijms, 2024, doi:10.3390/ijms25189939_

Round 1

Reviewer 1 Report

Comments and Suggestions for Authors

My comments are as follows:

Major comments

1 The first observation is that there is nothing new in this research, as the effect of decursin on colon cancer cells has been studied since 2010 in the same cells and in the animal model.

-Treatment of HCT116 colon cancer cells with decursin resulted in G1-phase cell cycle arrest, as revealed by FACS analyses. In addition, decursin increased protein levels of p21WAF1 with a decrease in cyclins and cyclin dependent kinases (CDKs and apoptotic induction (Kim, et al., 2010)

-Furthermore, decursin suppressed the proliferation and invasion of CT-26 colon cancer cells along with lung metastases in mice by decreasing MMP-9 production via the ERK/JNK signaling pathway (Son et al., 2011).

-A recent study demonstrated that decursin inhibited the proliferation and EMT of HT29 and HCT116 colon cancer cells by downregulating N-cadherin and vimentin protein expression and upregulating E-cadherin and PI3K/AKT signaling pathway expression, thereby inhibiting metastasis (Yang et al., 2023).

2-The second note is that the authors did not study the toxicity of decursin on normal cells or in normal mice

Reference

1-Kim, W. J., Lee, S. J., Choi, Y. D., and Moon, S. K. (2010d). Decursin inhibits growth of human bladder and colon cancer cells via apoptosis, G1-phase cell cycle arrest and extracellular signal-regulated kinase activation. Int. J. Mol. Med. 25 (4), 635–641. doi:10.3892/ijmm_00000386

2-Son, S. H., Park, K. K., Park, S. K., Kim, Y. C., Kim, Y. S., Lee, S. K., et al. (2011). Decursin and decursinol from Angelica gigas inhibit the lung metastasis of murine colon carcinoma. Phytother. Res. 25 (7), 959–964. doi:10.1002/ptr.3372

3-Yang, Y., Hu, Y., Jiang, Y., Fu, X., and You, F. (2023). Decursin affects proliferation, apoptosis, and migration of colorectal cancer cells through PI3K/Akt signaling pathway. China J. Chin. Materia Medica 48 (09), 2334–2342. doi:10.19540/j.cnki.cjcmm.20230117.703

 Introduction

1-There are a lot of previous studies that dealt with the effect of decursin on colon cancer cells and the animal model. It was overlooked and should be mentioned here with an explanation of what is new in this research (Kim, et al., 2010, Son,

References need to be added

1-Kim, W. J., Lee, S. J., Choi, Y. D., and Moon, S. K. (2010). Decursin inhibits growth of human bladder and colon cancer cells via apoptosis, G1-phase cell cycle arrest and extracellular signal-regulated kinase activation. Int. J. Mol. Med. 25 (4), 635–641. doi:10.3892/ijmm_00000386

 2-Son, S. H., Park, K. K., Park, S. K., Kim, Y. C., Kim, Y. S., Lee, S. K., et al. (2011). Decursin and decursinol from Angelica gigas inhibit the lung metastasis of murine colon carcinoma. Phytother. Res. 25 (7), 959–964. doi:10.1002/ptr.3372 

3-Yang, Y., Hu, Y., Jiang, Y., Fu, X., and You, F. (2023). Decursin affects proliferation, apoptosis, and migration of colorectal cancer cells through PI3K/Akt signaling pathway. China J. Chin. Materia Medica 48 (09), 2334–2342. doi:10.19540/j.cnki.cjcmm.20230117.703

Methods

1-What is the number of animals used for each group?

2-The dosage used for requires a reference that specifies on what basis the dosage was chosen.

3-What method was used to sacrifice animals?

Comments on the Quality of English Language

In conclusion, it is important to write down the shortcomings of the study,

Author Response

Dear reviewers and editorial staffs in International Journal of Molecular Sciences

We greatly appreciate your thoughtful comments that improved our manuscript entitled “Decursin induces G1 cell cycle arrest and apoptosis through reactive oxygen species-mediated endoplasmic reticulum stress in human colorectal cancer cells in in vitro and xenograft models”, regarding the manuscript [IJMS-3200694]. Through the accurate comments made by the reviewers, we better understand the critical issues in this paper. We have revised the manuscript according to the Reviewer’s suggestions. We hope that our revised manuscript will be considered and accepted for publication in the International Journal of Molecular Sciences. We acknowledge that the scientific and clinical quality of our manuscript was improved by the scrutinizing efforts of the reviewers and editors. Point-by-point responses to the reviewers’ comments are provided below.

Thank you for your consideration. I look forward to hearing from you.

Sincerely,

Jeong-Ran Park, Ph.D.,

Division of Research Center,

Scripps Korea Antibody Institute,

Chuncheon 24341,

Republic of Korea

Comments and Suggestions for Authors

My comments are as follows:

Major comments

  1. The first observation is that there is nothing new in this research, as the effect of decursin on colon cancer cells has been studied since 2010 in the same cells and in the animal model.

-Treatment of HCT116 colon cancer cells with decursin resulted in G1-phase cell cycle arrest, as revealed by FACS analyses. In addition, decursin increased protein levels of p21WAF1 with a decrease in cyclins and cyclin dependent kinases (CDKs and apoptotic induction (Kim, et al., 2010)

-Furthermore, decursin suppressed the proliferation and invasion of CT-26 colon cancer cells along with lung metastases in mice by decreasing MMP-9 production via the ERK/JNK signaling pathway (Son et al., 2011).

-A recent study demonstrated that decursin inhibited the proliferation and EMT of HT29 and HCT116 colon cancer cells by downregulating N-cadherin and vimentin protein expression and upregulating E-cadherin and PI3K/AKT signaling pathway expression, thereby inhibiting metastasis (Yang et al., 2023).

Response to comment : We appreciate the reviewer’s valuable comments. As the reviewer stated, several studies have shown the anti-cancer effects of decursin on CRC. However, there have been no studies on its efficacy and mechanisms in human cells in vivo. Additionally, this is the first paper that suggests cell apoptosis in CRC occurs through ROS and ER stress. Therefore, based on reviewer’s valuable remarks, we have mentioned the differences from previous studies in the “Introduction” as follows (page 3, 65-87) : Decursin has been used in traditional folk medicine as a tonic and a remedy for anemia, colds, pain, and other ailments while AGN is also marketed internationally as a functional food for healthcare [17,18]. Decursin has been investigated to have a variety of therapeutic effects in increasing numbers of studies, such as anti-angiogenesis, anti-oxidative, and anti-inflammatory activities [19-22]. Notably, decursin has exhibited anti-cancer properties in multiple cancer types, such as prostate, breast, lung, bladder, and colon [20,23-25]. Some reports have demonstrated the effects of decursin on CRC. Son et al. showed that decursin suppress the proliferation and invasion of murine CT-26 colon carcinoma cells [26]. Kim et al. demonstrated that decursin inhibits cell growth in 253J bladder and HCT-116 colon cancer cells via cell cycle arrest and ERK activation in vitro [25]. Most recent reports have shown that decursin regulates epithelial-mesenchymal transition (EMT) via the PI3K/Akt signaling pathway in CRC cells [27]. These effects of decursin make it an attractive candidate for further investigation and development as a CRC treatment. However, the detailed molecular mechanisms underlying the anti-cancer activity of decursin in human CRC cells remain largely unexplored, particularly in in vivo models.

The present study represents the first comprehensive examination of the anti-cancer efficacy of decursin in the treatment of CRC cells, both in vitro and xenograft model, while also elucidating the underlying molecular mechanisms. In addition, this study aimed to investigate the biological activities of decursin in CRC using both in vitro and in vivo models, as well as to determine its underlying mechanism of action, providing scientific evidence regarding the therapeutic roles of decrusin.

  1. The second note is that the authors did not study the toxicity of decursin on normal cells or in normal mice

Reference

1-Kim, W. J., Lee, S. J., Choi, Y. D., and Moon, S. K. (2010d). Decursin inhibits growth of human bladder and colon cancer cells via apoptosis, G1-phase cell cycle arrest and extracellular signal-regulated kinase activation. Int. J. Mol. Med. 25 (4), 635–641. doi:10.3892/ijmm_00000386

2-Son, S. H., Park, K. K., Park, S. K., Kim, Y. C., Kim, Y. S., Lee, S. K., et al. (2011). Decursin and decursinol from Angelica gigas inhibit the lung metastasis of murine colon carcinoma. Phytother. Res. 25 (7), 959–964. doi:10.1002/ptr.3372

3-Yang, Y., Hu, Y., Jiang, Y., Fu, X., and You, F. (2023). Decursin affects proliferation, apoptosis, and migration of colorectal cancer cells through PI3K/Akt signaling pathway. China J. Chin. Materia Medica 48 (09), 2334–2342. doi:10.19540/j.cnki.cjcmm.20230117.703

▶Response to comment : We appreciate the reviewer’s constructive comments regarding the toxicity of decursin on normal cells or normal mice. Unfortunately, we did not explored the toxicity of decursin in normal colon cancer cells; however, there were no specific issues regarding body weight or other organs when decursin was administered in animal experiments. Additionally, AGN has been used as a traditional herb for centuries. Since AGN products are sold as functional foods for women's health in Europe and the United States, it appears that there are no safety concerns as long as they are consumed at appropriate dosages.

In addition, among the previous experimental studies, there are those that examined the toxicity of decursin in normal cells and animals, based on reviewer’s valuable comments, we have mentioned the toxicity of decursin on normal cells or normal mice to the relevant section of “Discussion” as follows (page 11, 237-251) : Phytochemicals are the primary source of biologically active compounds. Moreover, their nontoxic or less toxic nature to normal cells has gained attention from the scientific community and clinicians in the modern drug discovery field [52]. Several studies have examined the cytotoxicity of decursin in normal cells. Up to 100 µg/ml of AGN extract and 60 µM of decursin inhibit the growth of PANC-1 and MIA PaCa-2 pancreatic cancer cells, but not normal pancreatic cells [53]. Decursin at 80 µM exhibited an apoptotic effect in human KBM-5 leukemia cells but did not show cytotoxicity against human peripheral blood lymphocytes at concentrations up to 80 µM [54]. Furthermore, in the non-neoplastic human prostate epithelial PWR-1E cells, decursin treatment did not lead to any increase in dead cells after 72 h of treatment at a dose of 100 μM [55]. Additionally, Kim et al. evaluated the acute and subacute effects of decursin in SD rats to determine its effects on body weight, histopathological, biochemical, and hematochemical changes. There were no significant differences in body weight, hematology, and biochemical parameters between the control and treated groups [56]. These results suggest that decursin may be an effective and safe treatment option for CRC therapy.

Introduction

  1. There are a lot of previous studies that dealt with the effect of decursin on colon cancer cells and the animal model. It was overlooked and should be mentioned here with an explanation of what is new in this research.

References need to be added

1-Kim, W. J., Lee, S. J., Choi, Y. D., and Moon, S. K. (2010). Decursin inhibits growth of human bladder and colon cancer cells via apoptosis, G1-phase cell cycle arrest and extracellular signal-regulated kinase activation. Int. J. Mol. Med. 25 (4), 635–641. doi:10.3892/ijmm_00000386 [25]

2-Son, S. H., Park, K. K., Park, S. K., Kim, Y. C., Kim, Y. S., Lee, S. K., et al. (2011). Decursin and decursinol from Angelica gigas inhibit the lung metastasis of murine colon carcinoma. Phytother. Res. 25 (7), 959–964. doi:10.1002/ptr.3372 [26]

3-Yang, Y., Hu, Y., Jiang, Y., Fu, X., and You, F. (2023). Decursin affects proliferation, apoptosis, and migration of colorectal cancer cells through PI3K/Akt signaling pathway. China J. Chin. Materia Medica 48 (09), 2334–2342. doi:10.19540/j.cnki.cjcmm.20230117.703 [27]

▶Response to comment : Based on the reviewer’s valuable remarks, we have added research on the effects of decursin on previously studied colon cancer cells and animal models. Additionally, we have described the new aspects of this study that differ from previous research This has been added to the relevant section of the 'Introduction' as follows (page 3, 65-88) : Decursin is a bioactive phytochemical originally isolated from Angelica gigas Nakai (AGN) and one of the most popular traditional medicines in East Asian countries. It has been used in traditional folk medicine as a tonic and a remedy for anemia, colds, pain, and other ailments while AGN is also marketed internationally as a functional food for healthcare[17,18]. Decursin has been investigated to have a variety of therapeutic effects in increasing numbers of studies, such as anti-angiogenesis, anti-oxidative, and anti-inflammatory activities [19-22]. Notably, decursin has exhibited anti-cancer properties in multiple cancer types, such as prostate, breast, lung, bladder, and colon [20,23-25]. Some reports have demonstrated the effects of decursin on CRC. Son et al. showed that decursin suppress the proliferation and invasion of murine CT-26 colon carcinoma cells [26]. Kim et al. demonstrated that decursin inhibits cell growth in 253J bladder and HCT-116 colon cancer cells via cell cycle arrest and ERK activation in vitro [25]. Most recent reports have shown that decursin regulates epithelial-mesenchymal transition (EMT) via the PI3K/Akt signaling pathway in CRC cells [27]. These effects of decursin make it an attractive candidate for further investigation and development as a CRC treatment. However, the detailed molecular mechanisms underlying the anti-cancer activity of decursin in CRC cells remain largely unexplored. Additionally, little is known about the systemic exposure of decursin following administration, as its anti-cancer activity has not been well explored in vivo compared to its in vitro effects.

The present study represents the first comprehensive examination of the anti-cancer efficacy of decursin in the treatment of CRC cells, both in vitro and xenograft model, while also elucidating the underlying molecular mechanisms. In addition, this study aimed to investigate the biological activities of decursin in CRC using both in vitro and in vivo models, as well as to determine its underlying mechanism of action, providing scientific evidence regarding the therapeutic roles of decrusin.

Methods

1-What is the number of animals used for each group?

▶Response to comment : Text added to the “Materials and Methods” section : (page 14, 329-331) When the tumors reached approximately 75–100 mm3, the mice were randomly divided into a control (PBS) and a decursin group (10 mg/kg), with five mice per group.

2-The dosage used for requires a reference that specifies on what basis the dosage was chosen.

▶Response to comment : We added the references to the “Discussion” section : (page 9, 191-197) Most of the presently available anti-cancer drugs have the ability to induce tumor cell apoptosis. In various cancer cells, anti-cancer effects of decursin have been documented: gastric cancer cells ( IC50 = 50 µM for 48 h) [38], bladder cancer cells ( IC50 = 50 µM for 24 h) [25], melanoma ( IC50 = 80 µM for 48 h) [21], and multiple myeloma cells ( IC50 = 80 µM for 24 h – 48 h) [39]. In the present study, decursin was also able to effectively inhibit the cell growh of HCT-116 and HCT-8 CRC cells at 48 h, with IC50 values of 50.33 µM and 49.68 µM, respectively.

3-What method was used to sacrifice animals?

▶ Response to comment : We have described the method of sacrificing mice to the “Materials and Methods” section : (page 14, 327-329) All mice were anesthetized using isoflurane inhalation (Hana Pharma, Gyeonggi-do, Republic of Korea) and euthanized with CO2 at the end of the experiment, after which the tumors were harvested for further analysis.

Reviewer 2 Report

Comments and Suggestions for Authors

Kim D et al., presented the work on Decursin on G1 cell cycle arrest in CRC. The findings suggest that Decursin induces cell cycle arrest and apoptosis in human CRC cells via ROS-mediated ER stress, suggesting that Decursin could be a therapeutic agent for CRC.

The cell viability assay in the two cell lines looks very close to each other (Figure 1A). These two cell lines are quite similar. It could have been more interesting if the author could use another cell line as a negative control.

It will be also helpful to present the quantification of Figure 1C.

Author Response

Dear reviewers and editorial staffs in International Journal of Molecular Sciences

We greatly appreciate your thoughtful comments that improved our manuscript entitled “Decursin induces G1 cell cycle arrest and apoptosis through reactive oxygen species-mediated endoplasmic reticulum stress in human colorectal cancer cells in in vitro and xenograft models”, regarding the manuscript [IJMS-3200694]. Through the accurate comments made by the reviewers, we better understand the critical issues in this paper. We have revised the manuscript according to the Reviewer’s suggestions. We hope that our revised manuscript will be considered and accepted for publication in the International Journal of Molecular Sciences. We acknowledge that the scientific and clinical quality of our manuscript was improved by the scrutinizing efforts of the reviewers and editors. Point-by-point responses to the reviewers’ comments are provided below.

Thank you for your consideration. I look forward to hearing from you.

Sincerely,

Jeong-Ran Park, Ph.D.,

Division of Research Center,

Scripps Korea Antibody Institute,

Chuncheon 24341,

Republic of Korea

Comments and Suggestions for Authors

Manuscript ID- IJMS-3200694

Kim D et al., presented the work on Decursin on G1 cell cycle arrest in CRC. The findings suggest that Decursin induces cell cycle arrest and apoptosis in human CRC cells via ROS-mediated ER stress, suggesting that Decursin could be a therapeutic agent for CRC.

The cell viability assay in the two cell lines looks very close to each other (Figure 1A). These two cell lines are quite similar. It could have been more interesting if the author could use another cell line as a negative control.

It will be also helpful to present the quantification of Figure 1C.

Response to comment : We appreciate the reviewer’s comments. HCT-116 and HCT-8 CRC cells possess KRAS mutations, which are commonly found in CRC and affect cell growth and response to anticancer therapy. Therefore, we explored the anticancer activity of decursin on both cell lines. Additionally, to further analyze the effects of decursin treatment on cell viability in CRC cells, decursin was administered for 24 to 72 hours. We have mentioned cell viability results to the relevant section of “Results” (page 5, 91-99) : To investigate the effects of decursin on the cell growth of CRC cells in vitro, the viability rate of HCT-116 and HCT-8 CRC cells was assessed by MTT assay. Treatment with decursin inhibited cell viability in a dose- and time- dependent manner at different concentrations (3.125, 6.25, 12.5, 25, 50, 100, and 200 µM) for 24 to 72 h compared with control cells (Figure 1A). In addition, the half-maximal inhibitory concentration (IC50) value for 72 h of decursin treatment was approximately 35 µM in HCT-116 and 30 µM in HCT-8 CRC cells (Figure 1B). These values indicate that HCT-116 and HCT-8 CRC cells are sensitive to decursin treatment. Based on the IC50 measured in the study, concentrations of 50 and 100 μM were selected for use in the following experiments.

Figure 1. Please see the attach.

Reviewer 3 Report

Comments and Suggestions for Authors

Manuscript ID- IJMS-3200694

            The paper entitled “Decursin induces G1 cell cycle arrest and apoptosis through reactive oxygen species-mediated endoplasmic reticulum stress in human colorectal cancer cells in in vitro and xenograft models” deals with the in-vitro and in vivo analysis of decursin towards cell cycle arrest induction and apoptosis in human CRC cells via ROS-mediated ER stress. This works showed that natural coumarin decursin could be a therapeutic agent for CRC.

Questions-

1.    What is the source of decursin used in this study, and what was its purity? I didn’t find any mention in the material and methods section. Was it purchased or isolated from natural source?  How was it dissolved, what solvent was used? All this information is missing

2.    What is the significance of studying decursin's effects on colorectal cancer? Is it solely on the basis of side effects of currently available treatment options for colorectal cancer?

Apart from two references cited in the manuscript, Ref 24 and 25, are no other studies available? What was the IC50 concentration in these two papers. How well does the current result align with these previous studies? This part needs to be discussed in the discussion section inspite of the difference in the cell types.

3.    Why the cytotoxicity at time point of 72h was not provided or studied, some reasoning should be provided since the IC50 values observed at 48h are on the higher side (~ 50 µM) for both cell types.

4.    What is the toxicity profile of decursin in normal cells versus cancer cells? Except for HCT-116 and HCT-8 CRC cells, data of the effect of decursin on normal cell was not studied. Any particular reasoning for not providing this crucial data should be explained?

5.    In the opinion of the authors what could be the potential side effects or limitations of using decursin as a cancer treatment option?

6.    What are the possible metabolites of decursin (either in-vitro or in-vivo) ? In addition, would any of them be more potent than decursin? Some SAR discussion to shed light on this effect is necessary in context to use of decursin in potential therapy of CRC.

7.    What further studies are needed to confirm decursin's therapeutic potential in CRC ? suggest a future plan of action.

8.    Could decursin be used in combination with existing cancer treatments? Please provide your opinions.

Author Response

Dear reviewers and editorial staffs in International Journal of Molecular Sciences

We greatly appreciate your thoughtful comments that improved our manuscript entitled “Decursin induces G1 cell cycle arrest and apoptosis through reactive oxygen species-mediated endoplasmic reticulum stress in human colorectal cancer cells in in vitro and xenograft models”, regarding the manuscript [IJMS-3200694]. Through the accurate comments made by the reviewers, we better understand the critical issues in this paper. We have revised the manuscript according to the Reviewer’s suggestions. We hope that our revised manuscript will be considered and accepted for publication in the International Journal of Molecular Sciences. We acknowledge that the scientific and clinical quality of our manuscript was improved by the scrutinizing efforts of the reviewers and editors. Point-by-point responses to the reviewers’ comments are provided below.

Thank you for your consideration. I look forward to hearing from you.

Sincerely,

Jeong-Ran Park, Ph.D.,

Division of Research Center,

Scripps Korea Antibody Institute,

Chuncheon 24341,

Republic of Korea

Comments and Suggestions for Authors

Manuscript ID- IJMS-3200694

The paper entitled “Decursin induces G1 cell cycle arrest and apoptosis through reactive oxygen species-mediated endoplasmic reticulum stress in human colorectal cancer cells in in vitro and xenograft models” deals with the in-vitro and in vivo analysis of decursin towards cell cycle arrest induction and apoptosis in human CRC cells via ROS-mediated ER stress. This works showed that natural coumarin decursin could be a therapeutic agent for CRC.

Questions-

  1. What is the source of decursin used in this study, and what was its purity? I didn’t find any mention in the material and methods section. Was it purchased or isolated from natural source? How was it dissolved, what solvent was used? All this information is missing

Response to comment : Thanks for the kind comment. We added source of decursin in “Materials and Methods” section as follows (page 12, 266-268): Decursin (99.99% purity, Selleckchem, Inc., Houston, TX, USA) was dissolved in DMSO, to prepare a stock solution. The final concentration of DMSO in treatment solutions was less than 0.1%.

  1. What is the significance of studying decursin's effects on colorectal cancer? Is it solely on the basis of side effects of currently available treatment options for colorectal cancer?

Response to comment : We appreciate the reviewer’s valuable comments. As the reviewer stated, several studies have shown the anti-cancer effects of decursin on CRC. However, there have been no studies on its efficacy and mechanisms in human cells in vivo. Additionally, this is the first paper that suggests cell apoptosis in CRC occurs through ROS and ER stress. Therefore, based on reviewer’s valuable remarks, we have mentioned the differences from previous studies in the “Introduction” as follows (page 3, 65-88) : Decursin has been used in traditional folk medicine as a tonic and a remedy for anemia, colds, pain, and other ailments while AGN is also marketed internationally as a functional food for healthcare [17,18]. Decursin has been investigated to have a variety of therapeutic effects in increasing numbers of studies, such as anti-angiogenesis, anti-oxidative, and anti-inflammatory activities [19-22]. Notably, decursin has exhibited anti-cancer properties in multiple cancer types, such as prostate, breast, lung, bladder, and colon [20,23-25]. Some reports have demonstrated the effects of decursin on CRC. Son et al. showed that decursin suppress the proliferation and invasion of murine CT-26 colon carcinoma cells [26]. Kim et al. demonstrated that decursin inhibits cell growth in 253J bladder and HCT-116 colon cancer cells via cell cycle arrest and ERK activation in vitro [25]. Most recent reports have shown that decursin regulates epithelial-mesenchymal transition (EMT) via the PI3K/Akt signaling pathway in CRC cells [27]. These effects of decursin make it an attractive candidate for further investigation and development as a CRC treatment. However, the detailed molecular mechanisms underlying the anti-cancer activity of decursin in CRC cells remain largely unexplored. Additionally, little is known about the systemic exposure of decursin following administration, as its anti-cancer activity has not been well explored in vivo compared to its in vitro effects.

The present study represents the first comprehensive examination of the anti-cancer efficacy of decursin in the treatment of CRC cells, both in vitro and xenograft model, while also elucidating the underlying molecular mechanisms. In addition, this study aimed to investigate the biological activities of decursin in CRC using both in vitro and in vivo models, as well as to determine its underlying mechanism of action, providing scientific evidence regarding the therapeutic roles of decrusin.

Apart from two references cited in the manuscript, Ref 24 and 25, are no other studies available? What was the IC50 concentration in these two papers. How well does the current result align with these previous studies? This part needs to be discussed in the discussion section inspite of the difference in the cell types.

▶Response to comment : As suggested by the reviewer, we have described the effects of decursin on the IC50 values in other cancer cells in the “Discussion” section (page 9, 191-197) : Most of the presently available anti-cancer drugs have the ability to induce tumor cell apoptosis. In various cancer cells, anti-cancer effects of decursin have been documented: gastric cancer cells ( IC50 = 50 µM for 48 h) [38], bladder cancer cells ( IC50 = 50 µM for 24 h) [25], melanoma ( IC50 = 80 µM for 48 h) [21], and multiple myeloma cells ( IC50 = 80 µM for 24 h – 48 h) [39]. In the present study, decursin was also able to effectively inhibit the cell growh of HCT-116 and HCT-8 CRC cells at 48 h, with IC50 values of 50.33 µM and 49.68 µM, respectively.

  1. Why the cytotoxicity at time point of 72h was not provided or studied, some reasoning should be provided since the IC50 values observed at 48h are on the higher side (~ 50 µM) for both cell types.

▶Response to comment : We thank the reviewer for this suggestion and have conducted a cell cytotoxicity assay of decursin on CRC cells for 24 to 72 hours, and we have included these data in the manuscript.

Text added to “Results” section (page 5, 92-100) : To investigate the effects of decursin on the cell growth of CRC cells in vitro, the viability rate of HCT-116 and HCT-8 CRC cells was assessed by MTT assay. Treatment with decursin inhibited cell viability in a dose- and time- dependent manner at different concentrations (3.125, 6.25, 12.5, 25, 50, 100, and 200 µM) for 24 to 72 h compared with control cells (Figure 1A). In addition, the half-maximal inhibitory concentration (IC50) value for 72 h of decursin treatment was approximately 35 µM in HCT-116 and 30 µM in HCT-8 CRC cells (Figure 1B). These values indicate that HCT-116 and HCT-8 CRC cells are sensitive to decursin treatment. Based on the IC50 measured in the study, concentrations of 50 and 100 μM were selected for use in the following experiments.

Figure 1.

  1. What is the toxicity profile of decursin in normal cells versus cancer cells? Except for HCT-116 and HCT-8 CRC cells, data of the effect of decursin on normal cell was not studied. Any particular reasoning for not providing this crucial data should be explained?

▶Response to comment : We appreciate the reviewer’s constructive comments regarding the toxicity of decursin on normal cells or normal mice. Unfortunately, we did not explored the toxicity of decursin in normal colon cancer cells; however, there were no specific issues regarding body weight or other organs when decursin was administered in animal experiments. Additionally, AGN has been used as a traditional herb for centuries. Since AGN products are sold as functional foods for women's health in Europe and the United States, it appears that there are no safety concerns as long as they are consumed at appropriate dosages.

In addition, among the previous experimental studies, there are those that examined the toxicity of decursin in normal cells and animals, based on reviewer’s valuable comments, we have mentioned the toxicity of decursin on normal cells or normal mice to the relevant section of “Discussion” as follows (page 11, 237-251) : Phytochemicals are the primary source of biologically active compounds. Moreover, their nontoxic or less toxic nature to normal cells has gained attention from the scientific community and clinicians in the modern drug discovery field [52]. Several studies have examined the cytotoxicity of decursin in normal cells. Up to 100 µg/ml of AGN extract and 60 µM of decursin inhibit the growth of PANC-1 and MIA PaCa-2 pancreatic cancer cells, but not normal pancreatic cells [53]. Decursin at 80 µM exhibited an apoptotic effect in human KBM-5 leukemia cells but did not show cytotoxicity against human peripheral blood lymphocytes at concentrations up to 80 µM [54]. Furthermore, in the non-neoplastic human prostate epithelial PWR-1E cells, decursin treatment did not lead to any increase in dead cells after 72 h of treatment at a dose of 100 μM [55]. Additionally, Kim et al. evaluated the acute and subacute effects of decursin in SD rats to determine its effects on body weight, histopathological, biochemical, and hematochemical changes. There were no significant differences in body weight, hematology, and biochemical parameters between the control and treated groups [56]. These results suggest that decursin may be an effective and safe treatment option for CRC therapy.

  1. In the opinion of the authors what could be the potential side effects or limitations of using decursin as a cancer treatment option?

▶Response to comment : The safety of decursin has primarily been demonstrated through in vivo studies conducted on rodents. Pharmacokinetic studies are also predominantly based on rodent research, with limited investigations involving humans(1-2). Currently, there is insufficient data to support its clinical application, and to use decursin as an anticancer agent, it is essential to conduct various clinical trials to enhance the drug's reliability. Additionally, there is a lack of research on decursin's drug resistance and off-target toxicity. Further studies are warranted to address these issues, including efforts to overcome multidrug resistance and improve drug structure.

References

  • Kim, Sook-Jin, et al. "Simultaneous determination of decursin, decursinol angelate, nodakenin, and decursinol of angelica gigas nakai in human plasma by UHPLC-MS/MS: application to pharmacokinetic study." Molecules5 (2018): 1019.
  • Zhang, Jinhui, et al. "Single oral dose pharmacokinetics of decursin and decursinol angelate in healthy adult men and women." PLoS One2 (2015): e0114992.

  1. What are the possible metabolites of decursin (either in-vitro or in-vivo) ? In addition, would any of them be more potent than decursin? Some SAR discussion to shed light on this effect is necessary in context to use of decursin in potential therapy of CRC.

▶Response to comment : Decursin undergoes metabolic transformation in the body, leading to the formation of various metabolites such as decursinol and decursinol angelate. Metabolic analyses of decursin and decursinol angelate were conducted using human liver microsomes and rodent plasma, which showed that decursinol is produced as a major metabolite of decursin (1,2). Decursinol and decursinol angelate exhibit significant anti-cancer properties, with potential applications in cancer therapy. However, decursinol is reported to be one of the main active metabolites of decursin, and some studies indicate that this metabolite has lower biological activity compared to its parent compound, decursin (3,4). Decursinol angelate is a structural analogue of decursin, and it has been reported to significantly enhance biological activities compared to decursinol. It shows potent anti-cancer effects and may influence various cell signaling pathways, contributing to its potential therapeutic applications (5,6). Further research is needed to fully understand the mechanisms and potential applications of decursin and its metabolites.

According to Structure-Activity Relationship (SAR) studies on decursin and its metabolites, the coumarin ring and the oxopropyl group in the parent structure play important roles in their biological activity. Decursin itself exhibits relatively strong anticancer activity; however, if the coumarin ring is disrupted or structurally altered during the metabolic process, its activity may decrease. Conversely, structural modifications in the active site could enhance stability and broaden the range of indications (7-9). Although we are not experts in SAR studies, we believe that it is important to establish or develop therapeutic strategies that consider the metabolites of decursin through such SAR research. This could lead to the development of effective treatments for diseases such as CRC.

Rerferences

  • Li L, Zhang JH, Xing CG, Kim SH, Jiang C, Lu J. In Vitro Metabolism of Pyranocoumarin Isomers Decursin and Decursinol Angelate by Liver Microsomes from Man and Rodent. Planta Medica 2013; 79(16): 1536–1544.
  • Anticancer potential of decursin, decursinol angelate, and decursinol from Angelica gigas Nakai: A comprehensive review and future therapeutic prospects
  • Song, Gyu-Yong, et al. "Decursin suppresses human androgen-independent PC3 prostate cancer cell proliferation by promoting the degradation of β-catenin." Molecular Pharmacology6 (2007): 1599-1606.
  • im, Dongsool, et al. "A novel anticancer agent, decursin, induces G1 arrest and apoptosis in human prostate carcinoma cells." Cancer research3 (2005): 1035-1044.
  • Kim, Won-Jung, et al. "Decursinol angelate blocks transmigration and inflammatory activation of cancer cells through inhibition of PI3K, ERK and NF-κB activation." Cancer letters1 (2010): 35-42.
  • Chang, Sukkum Ngullie, et al. "Decursinol angelate arrest melanoma cell proliferation by initiating cell death and tumor shrinkage via induction of apoptosis." International journal of molecular sciences8 (2021): 4096.
  • Yim, Dongsool, et al. "A novel anticancer agent, decursin, induces G1 arrest and apoptosis in human prostate carcinoma cells." Cancer research3 (2005): 1035-1044.
  • Ali, Md Yousof, et al. "Angiotensin-I-converting enzyme inhibitory activity of coumarins from Angelica decursiva." Molecules21 (2019): 3937.
  • Lee, Wonhwa, et al. "JH‐4 reduces HMGB1‐mediated septic responses and improves survival rate in septic mice." Journal of Cellular Biochemistry4 (2019): 6277-6289.
  1. What further studies are needed to confirm decursin's therapeutic potential in CRC ? suggest a future plan of action.

▶Response to comment : To utilize decursin in clinical practice, further studies are required to evaluate its efficacy and toxicity in combination with currently used anticancer agents such as 5-FU and oxaliplatin, as well as to investigate its potential for improving drug resistance. Actually, we have explored the potential therapeutic effects of decursin in combination with oxaliplatin, a known active agent in first-line chemotherapy for advanced CRC. The results indicate that decursin sensitizes anticancer agents (data not shown), and further studies are currently ongoing.

  1. Could decursin be used in combination with existing cancer treatments? Please provide your opinions.

▶Response to comment : Various preclinical and clinical studies revealed that natural products and their combinations with chemotherapeutics mediate their anticancer effects via modulation of various signaling pathways implicated in reducin cell proliferation and promoting apoptosis and cell cycle arrest. Therfore, we have explored the potential therapeutic effects of decursin in combination with oxaliplatin, a known active agent in first-line chemotherapy for advanced CRC. The results indicate that decursin sensitizes anticancer agents (data not shown), and further studies are currently ongoing. While additional research is necessary, these findings provide a positive outlook on the use of decursin in conjunction with existing cancer therapies.

Text added to “Discussion” section (page 11, 251-257) : Phytochemicals have been used for the treatment and prevention of various diseases, either alone or in combination with other drugs [57, 58]. Therefore, we have explored the potential therapeutic effects of decursin in combination with oxaliplatin, a known active agent in first-line chemotherapy for advanced CRC. The results indicate that decursin sensitizes anticancer agents (data not shown), and further studies are currently ongoing. Additional detailed preclinical research is required to identify and discover more effective synergistic combinations of decursin with alternative medications to prevent the failure of CRC therapy.

Round 2

Reviewer 3 Report

Comments and Suggestions for Authors

Comments to authors-

I am very pleased to note that the authors have greatly improved the manuscript in terms of scientific quality. I was also pleased to see that the IC50 values at time point of 72 h has improved.

I agree that AGN has no toxicity when administered at appropriate doses, but for the sake of toxicity evaluation, it is always a regulatory requirement to present the data at the time of co-assessment in cancer cells. Therefore, I had mentioned the same in my evaluation. For the current manuscript, the reasoning and literature provided by the authors is sufficient. However, I would suggest generating your own in-house data available for your future reference.

Check spelling @ last line of introduction “decrusin” to “decursin.”

Include DOI for all the references in the reference section.